# Multi-Time and Multi-Band CSP Motor Imagery EEG Feature Classification Algorithm

**Jun Yang, Zhengmin Ma and Tao Shen \***

School of Information Engineering and Automation, Kunming University of Science and Technology, Kunming 650500, China; yang-jun@kust.edu.cn (J.Y.); zhengminma@stu.kust.edu.cn (Z.M.)
\* Correspondence: paradisewolf@126.com

**Abstract:** The effective decoding of motor imagination EEG signals depends on significant temporal, spatial, and frequency features. For example, the motor imagination of the single limbs is embodied in the μ (8–13 Hz) rhythm and β (13–30 Hz) rhythm in frequency features. However, the significant temporal features are not necessarily manifested in the whole motor imagination process. This paper proposes a Multi-Time and Frequency band Common Space Pattern (MTF-CSP)-based feature extraction and EEG decoding method. The MTF-CSP learns effective motor imagination features from a weak Electroencephalogram (EEG), extracts the most effective time and frequency features, and identifies the motor imagination patterns. Specifically, multiple sliding window signals are cropped from the original signals. The multi-frequency band Common Space Pattern (CSP) features extracted from each sliding window signal are fed into multiple Support Vector Machine (SVM) classifiers with the same parameters. The Effective Duration (ED) algorithm and the Average Score (AS) algorithm are proposed to identify the recognition results of multiple time windows. The proposed method is trained and evaluated on the EEG data of nine subjects in the 2008 BCI-2a competition dataset, including a train dataset and a test dataset collected in other sessions. As a result, the average cross-session recognition accuracy of 78.7% was obtained on nine subjects, with a sliding window length of 1 s, a step length of 0.4 s, and the six windows. Experimental results showed the proposed MTF-CSP outperforming the compared machine learning and CSP-based methods using the original signals or other features such as time-frequency picture features in terms of accuracy. Further, it is shown that the performance of the AS algorithm is significantly better than that of the Max Voting algorithm adopted in other studies.

**Keywords:** electroencephalogram decoding; motor imagery; common space pattern; sliding window



## 1. Introduction

Electroencephalograms (EEG) are a method used to record electrical information from the cerebral cortex, thus reflecting part of brain activity. The emotion, motor intention, health status, and other information of the subject can be identified by analyzing EEG signals [1–3]. Brain–computer interface (BCI) technology refers to the bridge of an information transmission channel between the human brain and external devices independent of the traditional neural center network of the brain, enabling the control of external devices through the human brain [4–7]. BCI has been increasingly applied in various applications such as motor rehabilitation, neural intervention, games, and entertainment [8,9]. However, EEG signals have limitations, such as low spatial resolution, low SNR, and non-stationarity. In addition, the collected EEG signals are often accompanied by artifact information. Thus, the acquisition of EEG is challenging, and available public data are limited [10,11].

To identify different EEG signals, machine learning-based methods have been proposed by researchers to identify different EEG signals. Machine learning is generally divided into two steps: feature extraction and classification. Common machine learning classification models are Linear Discriminant Analysis (LDA) [12], k-Nearest Neighbor

(KNN) [13], Support Vector Machines (SVM) [14], Kernel Naive Bayes [15], and so on. Machine learning methods have been used to extract distinctive biomarker features and to identify healthy people and stroke patients in [16–18]. Reference [19] proposed a Hybrid Machine Learning (HML) Classifier (including KNN, SVM, RF, NB, LR, CART, LDA, AB, GB, and ET) to perform bruxism detection. Ref. [20] generalized a fixed classification method for all subjects by combining the XDAWN spatial filter and the Riemannian Geometry Classifier (RGC) for P300-EEG signal decoding. In [21], the Common Spatial Patterns (CSP) and Linear Discriminant Analysis (LDA) machine learning algorithms were applied to EEG signals for feature extraction and classification, respectively.

Motor imagery (MI) EEG decoding aims to correctly analyze brain signal patterns (left and right hand, etc.), providing a basis for the implementation of an online BCI. In the MI experimental paradigm, selecting an effective period from a complete MI trial is a critical step. In most MI experimental paradigms, a complete 3–5 s-long trial is used as the smallest unit. However, on the one hand, the subject's attention and imagination ability is not necessarily a continued focus in the whole trial process. On the other hand, different subjects have different reaction times after receiving instructions. Therefore, it is a key problem to crop the most profitable MI period before decoding the data. Selecting adequate time windows and taking advantage of their characteristics can maximize the temporal features of MI-EEG signals. In addition to temporal features, EEG also includes frequency features and spatial features. The frequency features of MI-EEG are mainly manifested in μ rhythm (8–13 Hz) and β rhythm (13–30 Hz); that is, when performing unilateral limb MI, the signal energy in the corresponding frequency band of contralateral brain electrodes decreases [22–24].

Extracting effective MI features plays a vital role in the further identification of EEG signals. Many studies have been conducted to find the event-related synchronization (ERD) characteristics of contralateral brain electrodes on the frequency domain when performing unilateral limb MI [25–27]. Short-time Fourier transform (STFT) was explored to consider time and frequency features simultaneously, which converted the original signal into time-frequency domains [28,29].

The Common Space Pattern (CSP) algorithm has been widely studied due to its advantages in the extraction of significant MI-EEG features [30,31]. Many studies have used the CSP algorithm to extract significant EEG features in order to achieve good classification accuracy. In [32,33], the CSP algorithm was combined with the LDA classifier for MI-EEG decoding. In [34,35], the performance of five different features based on the CSP + SVM model was analyzed. Significant EEG signal features extracted by the CSP algorithm were able to improve the recognition accuracy. Many studies have improved CSP in order to obtain better EEG characteristics. Several studies have used different frequency bands to improve CSP, including sub-band CSP (SBCSP) [36,37], Filter Bank CSP (Filter Bank CSP, FBCSP [25,38], and Discriminative FBCSP (DFBCSP) [39].

The single frequency band and single time window method lose detail features and thus cannot fully reflect the complete features of MI-EEG. Despite some efforts to fuse different frequency characteristics, efficient and detailed time features have been ignored. The influence of the interception of different fixed time windows on the CSP algorithm has been explored [40]. However, the effective MI starting time and the effective time window length vary from person to person. Additionally, the generalization of a single fixed-length window method is challenging. Only a few studies have investigated an effective MI time window. In [41], an automatic selection of the best time window was proposed. In [42], the features of three time-windows were extracted based on a multi-band CSP algorithm.

For the simultaneous extraction of the effective time window and frequency features, this paper proposes a CSP algorithm based on multi-time window and multi-frequency band (MTF-CSP). Multiple Support Vector Machines (SVM) are employed to classify the multi-frequency band features extracted from multiple time windows. The ED and AS algorithm are used to make the final output decision. The proposed MTF-CSP methods were compared to state-of-the-art methods in terms of different extracted features, CSP-

based methods, and the final decision algorithms and performed better than previous methods. The classification accuracy of the two decision outputs obtained from two decision algorithms is compared, and the influence of different window lengths and numbers on the classification accuracy is analyzed. The major contributions of the paper are highlighted as follows:

- The MTF-CSP features extracted by us achieved better classification accuracy in the classification process by comparing it with the original signal and time-frequency features.
- The strategy for intercepting multiple sliding window EEG data for analysis demonstrated better performance than direct full-window EEG data analysis.
- We compared our MTF-CSP method with traditional CSP-based models, and the results demonstrated that the multi-band and multi-time strategy could obviously improve the recognition performance of the CSP-based models.
- For the final decision algorithm, we compared our model with the Max Voting method used in some studies [43], and the cross-session classification accuracy obtained using our proposed AS algorithm was significantly higher than that obtained by using Max Voting algorithm.

## 2. Materials and Methods

### 2.1. Data and Preprocessing

The experimental data were obtained from the BCI-IV-2a dataset in 2008 [44], which recorded four types of MI signals (including left hand, right hand, foot, and tongue MI tasks) from nine subjects (these subjects did not have any particular medical conditions according to the description of the data publisher). Figure 1 shows a MI experiment paradigm, where subjects wearing EEG caps sit in front of a computer screen. Before the start of the experiment, the computer screen shows a black cross symbol, and at the same time, the computer issues a prompt to remind the participant to pay attention. A second later, the screen gives the participant specific instructions that remind the participants to prepare themselves. At the third second, the subjects begin to perform the specified MI task. After 3 s, the subject is asked to rest for 1.5 s to prepare for the next task.

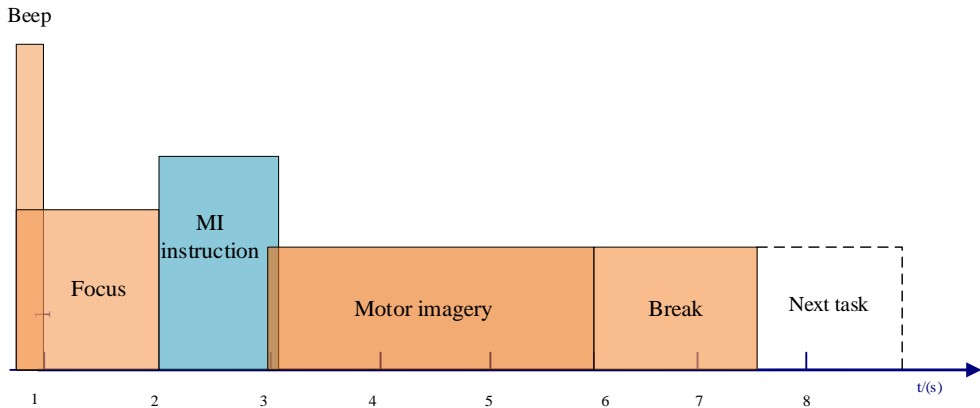

**Figure 1.** Schematic diagram of an MI experiment paradigm.

Each participant underwent two experimental sessions, and the two sessions were performed on different days. An EEG recording session was divided into 6 runs, and 48 trials were performed in each run. Therefore, each subject performed 288 training trials and 288 testing trials. That is, 288 training (including validation data) and 288 testing MI-EEG signals were captured from each subject. We used all of the subjects' EEG data from the two sessions for our research. Due to the limited amount of EEG data, all of the data collected from the first session were used for training and validation, and the data collected from the second session were used for cross-session testing.

For the MI-EEG trials for the left and right hands were 3 s long and were selected according to the labels and MI experiment paradigm for the following binary MI-EEG decoding work. All of these trials are first fit to 8–30 Hz, from which n sliding windows with a step length of s are cropped. Each window is then fir into two larger frequency bands, that is, 8–13 Hz (μ) and 13–30 Hz (β), which are again fit to two (8–10 Hz named μ-1, 10–13 Hz named μ-2) and three (13–18 Hz named β-1, 18–23 Hz named β-2, 23–30 Hz named β-3) smaller frequency bands.

## 2.2. MI-EEG Recognition Based on MTF-CSP

The proposed method consists of four parts: cropping sliding windows, MTF-CSP for feature extraction, multi-window SVM for classification, and acquiring the final decision for all of the windows; the logic flow diagram for the whole algorithm is shown in Figure 2.

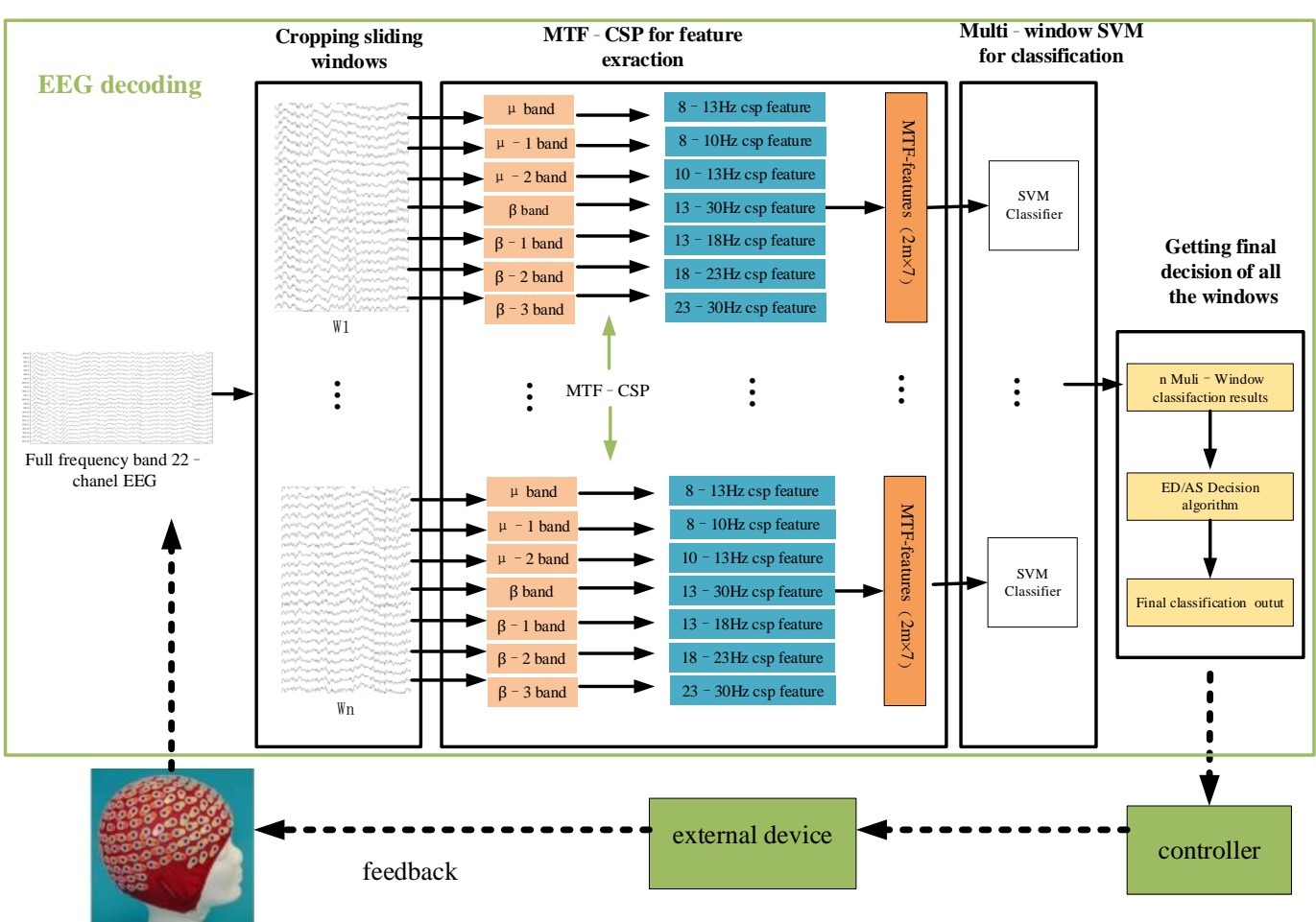

**Figure 2.** Logic flow diagram of the proposed method.

### 2.2.1. CSP Algorithm

A CSP algorithm is a spatial filtering algorithm that is suitable for processing multi-channel (multi-dimensional) signals such as EEG signals because CSP can synchronously exploit the spatial correlation of multiple signal channels to capture the spatial characteristics of the multi-dimensional signals. It aims to find an optimal set of spatial filters, enabling the two kinds of samples to acquire a spatial characteristic component and maximum variance after projection [33,45]. The process of achieving this optimal spatial filter is described by Formulas (1)–(4) as follows:

For the binary MI-EEG classification task, the training set data can be divided into two types of datasets: $X_1$ and $X_2$. The mixed covariance matrix ($R$) is obtained for the two types of data as follows:

$$R = \overline{R}_1 + \overline{R}_2 = \text{mean}(\frac{x_1 x_1^{\text{T}}}{\text{trace}(x_1 x_1^{\text{T}})}) + \text{mean}(\frac{x_2 x_2^{\text{T}}}{\text{trace}(x_2 x_2^{\text{T}})}) \qquad (1)$$

where $x_1$ and $x_2$ represent the samples in the two types of datasets, $X_1$ and $X_2$, respectively. $\overline{R}_1$ and $\overline{R}_2$ represent the average covariance matrix of the two types of data, respectively. The mean indicates the average of the two covariance matrices, and trace indicates the trace of the matrix.

The albino matrix is obtained according to the mixed covariance matrix:

$$P = \frac{1}{\sqrt{\lambda_{\text{r}}}} U_{\text{r}}^{\text{T}} \qquad (2)$$

where $\lambda_{\text{r}}$ and $U_{\text{r}}^{\text{T}}$ are the eigenvalue and eigenvector matrix of the mixed covariance matrix ($R$), respectively. By using the albino matrix and the mixed covariance matrix, the common eigenvector matrix $S_1$ and $S_1$ can be expressed as:

$$S_1 = P R_1 P^{\text{T}}, S_2 = P R_2 P^{\text{T}} \qquad (3)$$

The projection matrix is determined by the albino matrix ($P$) and the common eigenvector matrix ($S_1$ or $S_2$). The projection matrix ($W$) is expressed as:

$$W = U_{\text{s}}^{\text{T}} P \qquad (4)$$

where $U_{\text{s}}^{\text{T}}$ refers to the eigenvector matrix obtained by the decomposition of $S_1$ or $S_2$ ($S_1$ is equal to $S_2$).

The final CSP filter $W_{\text{CSP}}$ consists of m maximum values and m minimum values selected from the projection matrix ($W$). Finally, a feature vector with a length of 2 m is extracted. That is, the CSP feature vector of the single-frequency band and single-time window signal is $X_{\text{CSP}} = [x_1, \cdots, x_{2m}]$.

### 2.2.2. Multi-Time Window and Multi-Frequency Band CSP Strategy for Feature Extraction

This paper proposes a multi-time and multi-frequency CSP (MTF-CSP) to extract features for MI information. In the MTF-CSP, n time windows of length w are intercepted from the original signal of 3 s, and the interception step length is one s second. For a single time window signal, the sub-band time-amplitude signals of seven frequency bands (μ, μ-1, μ-2, β, β-1, β-2, β-3) are separated by inputting the original data into different band-pass filters. These seven frequency bands include the μ rhythm frequency band (8–13 Hz) and the β rhythm frequency band (13–30 Hz), with significant MI frequency features as well as the refined frequency bands (μ-1, μ-2, β, β-1, β-2, β-3) to simultaneously capture the global and local frequency features of the signal. The CSP algorithm was then used to extract the features of each frequency band for a single-time window signal in order to obtain a set of eigenvalues with the length of 2 m, which is a feature vector that maximizes the difference between the two classes of signals by mapping the signal to another space. The eigenvalues of the seven frequency bands are connected end to end to form a set of vectors that can be used as the features of a single MI-EEG signals in a single-time window. Thus, the features of the seven frequency bands filtered from a single window can finally combined into a vector with a length of 2 m × 7, as shown in Figure 3. The statistical results of the EEG features extracted from the seven frequency bands are also visualized in Figure 4 after dimension reduction takes place.

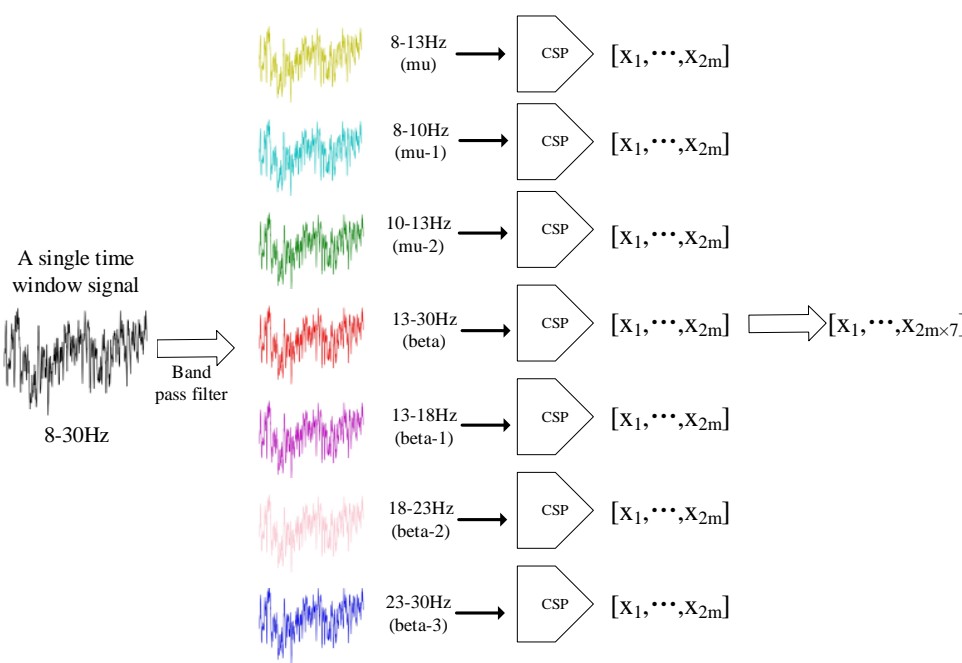

**Figure 3.** Multi–band features fusion based on CSP.

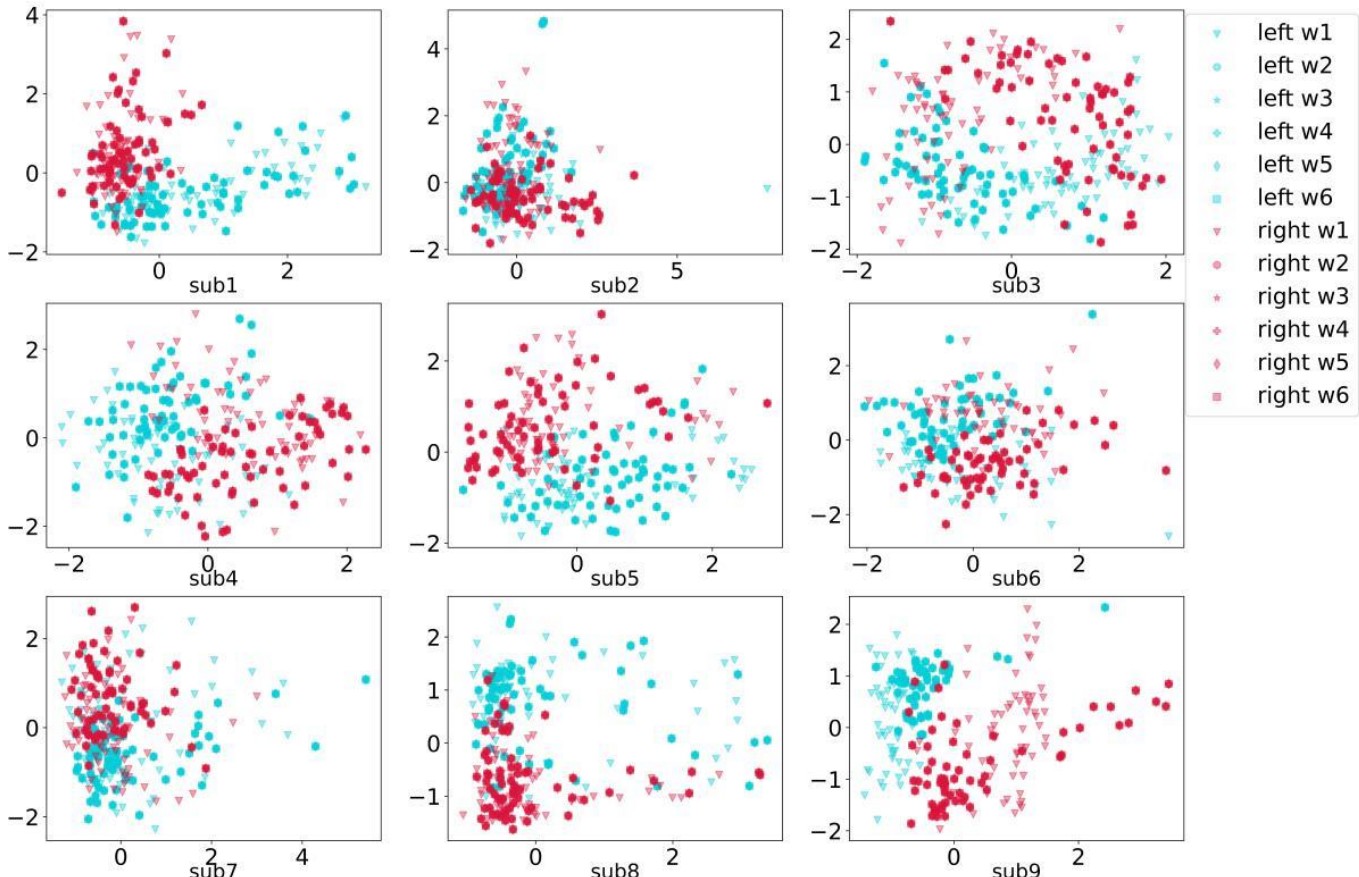

**Figure 4.** The statistical visualization scatter diagram of the multi-band CSP features of nine subjects.

### 2.2.3. SVM Classifier for Multi-Window EEG Classification

SVM is a supervised machine learning algorithm that can be used for classification or regression tasks. Before this, a linear classifier was used to find a hyperplane that could distinguish the two types of data. However, in many cases, there are countless similar hyperplanes. The purpose of SVM is to find the optimal hyperplane to distinguish the two types of samples. It is formulated to find an optimal hyperplane to distinguish the two categories and tries to maximize the margin between those categories [46,47]. The optimization objective is to find the hyperplane that is the farthest away from the support vector, which is the sample that is closest to the hyperplane that is decisive. The objective function is defined as follows:

$$\underset{w,b}{\mathrm{argmax}}\left\{ \frac{1}{\|w\|} \min_i \left[ y_i \cdot \left( w^\mathrm{T} \cdot \Phi(x_i) + b \right) \right] \right\} \tag{5}$$

where $w$ and $b$ are the decision surface parameters to be optimized for; $x_i$ and $y_i$ represent the ith sample and its label, respectively, and $\Phi$ is the kernel function, which maps the features of the samples to the higher dimensional space so as to transform the linearly indivisible problem in the lower dimensional space into a linearly separable problem in the higher dimensional space. In Formula (5), the subformula after min aims to find the vectors with the minimum distance from the hyperplane, namely the support vector, and the subformula after max aims to find the hyperplane with the maximum distance from the support vector. The Radial Basis Function (RBF) is used as the kernel function in this paper to map the EEG features to a higher dimension. The RBF function is represented by Formulas (6):

$$\Phi_\gamma(x) = \exp\left( -\gamma\|x - 1\|^2 \right) \tag{6}$$

where $x$ represents the data that need to be mapped into higher dimensions, $l$ represents the features of all of the samples, $\gamma$ is an adjustable parameter that represents the complexity of the transformation, exp computes exponential functions based on natural numbers, and $\|x - l\|^2$ represents the similarity between the data. The reasons why one may use the RBF function is to map low-dimensional features into higher-dimensional space by calculating the similarity values between a sample feature and all of other sample features as a new higher-dimensional vector.

The left- and right-hand MI features extracted by the MTF-CSP algorithm were classified by SVM. The multi-frequency band CSP features extracted from n time-window signals were fed into n SVM classifiers with the same parameters, obtaining n recognition values.

### 2.2.4. Final Decision over Multiple Time Windows

Two algorithms, ED and AS, are used to make the final recognition decision on the n recognition results obtained from multi-frequency CSP feature extraction and SVM classification in n time windows. They are described in the following.

1    Effective duration algorithm (ED)

The ED algorithm aims to determine the longest sequence with the successive same value in a list of digits as the effective duration of the sequence; this continuous value is determined as the final decision value. The decision output value has two characteristics: continuous and most frequent occurrences. That is, the final decision value should not only be sequential values in time order, and the number of consecutive occurrences is the highest compared to other values. The successive occurrences of multiple same identification results indicate the continuous and effective concentration of the subjects in the MI tasks. As shown in Figure 5, in the CSP-based feature extraction process, n groups of feature vectors (2 m × 7) are obtained that correspond to n time windows. These feature vectors are fed into n SVM classifiers with the same parameters, providing n recognition scores. The n results are arranged in order of time. The period with the largest number of same successive

decisions is regarded as containing more MI feature information (effective time period) in a single MI trial, and the same successive predicted value in this period is regarded as the final decision value. For example, in n time windows, if most of the consecutive identification results are for right-hand MI, then the model determines that this trial is a MI task for the right hand.

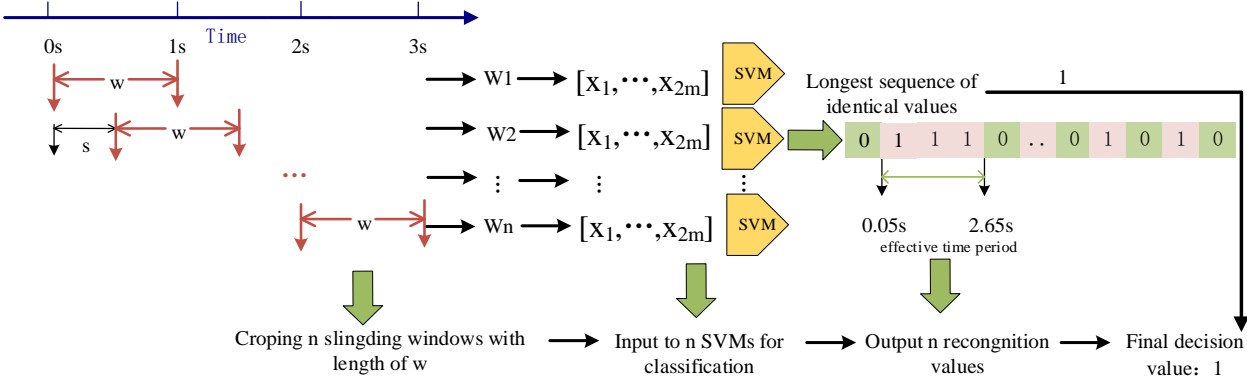

**Figure 5.** Schematic diagram of the ED decision algorithm.

2   Average score algorithm (AS)

In order to weigh the recognition results of each window fairly in terms of details, the AS algorithm, by which the scores of each time window obtained by SVM are added and averaged, is proposed. The core of the AS algorithm takes the SVM scores rather than the binary classification results of each time window as the reference value to make the final decision because these scores can specifically measure the category nature of the samples in detail. After calculating the average SVM scores of n time windows, the threshold is selected according to their precision and recall variation curves. The final decision is 1 (right hand MI) for scores higher than the threshold; otherwise, the final decision is 0 (left hand MI). SVM prediction score should be denoted in the *j*th window be $score_j$ and h should be the selected threshold value. Then, the final decision value $P$ is:

$$P = 1, mean\left(score_j\right) > \text{h}, (j = 1, 2, \ldots n)$$
$$P = 0, mean\left(score_j\right) \leq \text{h}, (j = 1, 2, \ldots n)$$

(7)

The decision process of the AS algorithm is depicted in Figure 6.

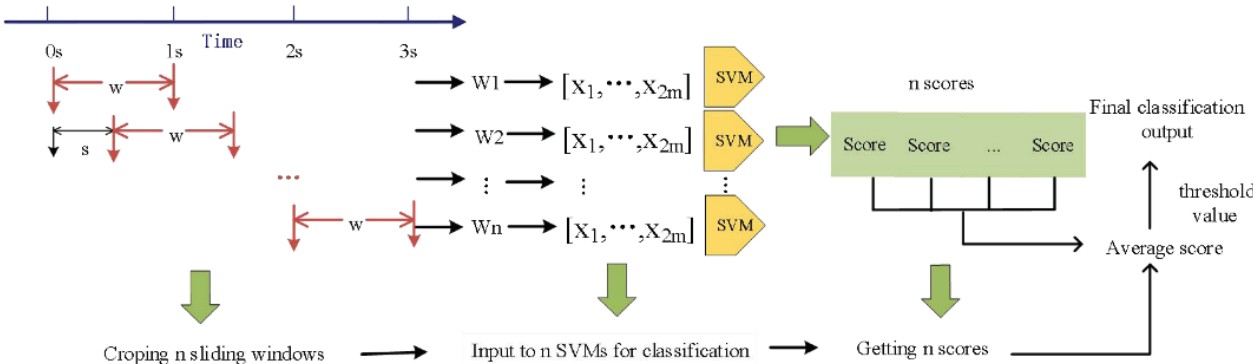

**Figure 6.** Schematic diagram of the AS decision algorithm.

Since the recall rate and precision vary according to the threshold [48,49], an appropriate threshold should be determined to judge the final score. The true positive (*TP*) denotes the number of positive samples that have been determined correctly, and the true negative (*TN*) denotes the number of negative samples that have been determined correctly. The

false positive (*FP*) and the false negative (*FN*) denote the numbers of positive samples and negative samples determined falsely, respectively. The recall *r* and precision *p* are defined as follows:

$$r = \frac{TP}{TP + FN}, \\ p = \frac{TP}{TP + FP} \tag{8}$$

Figure 7 shows the recall rate and precision curves of nine subjects according to a varied threshold. As shown in Figure 7, the recall rate and precision change with opposite trends. In order to weigh the recall rate and precision, the threshold value at the intersection point of the two curves is taken as the optimal threshold value for classification.

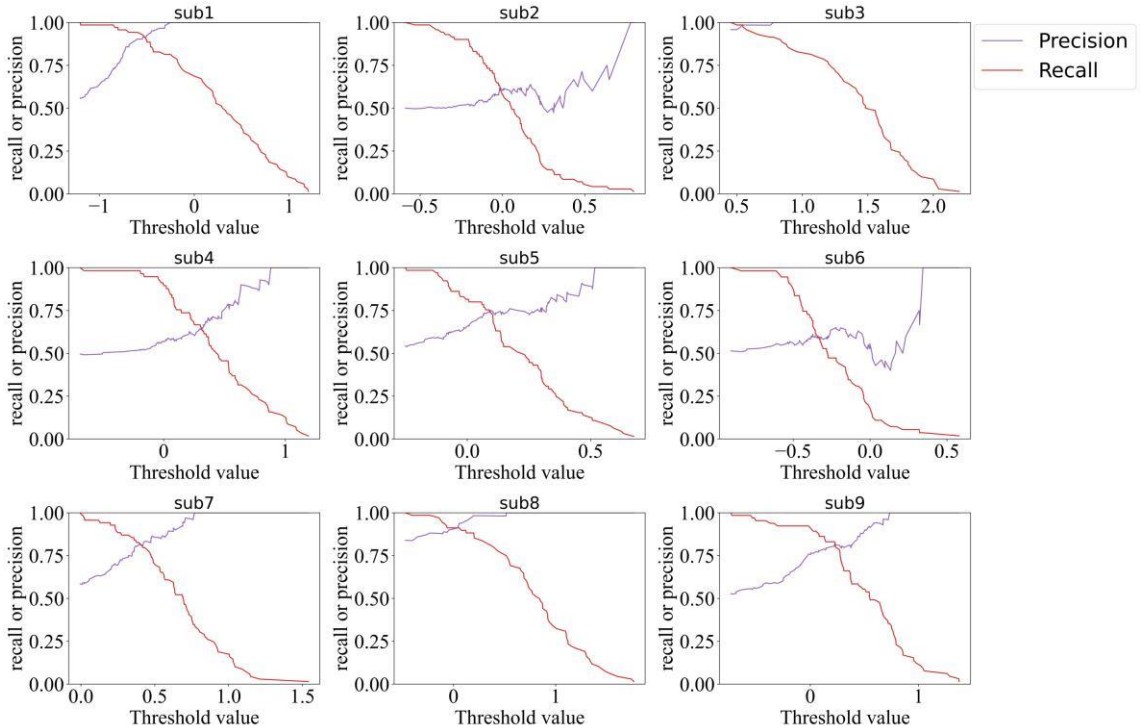

**Figure 7.** The precision and recall curves on nine subjects for selecting the optimal threshold value.

## 3. Results

The model was trained on the training dataset, tested on the test dataset collected from another session, and validated using the 4-fold cross-validation method. The validation accuracy is demonstrated in Tables 1 and 2.

**Table 1.** The validation accuracy for ED algorithm.

|  | 6 Windows | | | | 11 Windows | | | |
|---|---|---|---|---|---|---|---|---|
|  | 1 s | 1.5 s | 2 s | 2.5 s | 1 s | 1.5 s | 2 s | 2.5 s |
| Sub1 | 0.986 | 0.920 | 0.928 | 0.906 | 0.978 | 0.934 | 0.913 | 0.906 |
| Sub2 | 0.956 | 0.971 | 0.926 | 0.875 | 0.993 | 0.963 | 0.897 | 0.882 |
| Sub3 | 0.957 | 0.942 | 0.934 | 0.934 | 0.964 | 0.942 | 0.927 | 0.927 |
| Sub4 | 0.953 | 0.977 | 0.953 | 0.961 | 0.969 | 0.946 | 0.961 | 0.961 |
| Sub5 | 0.961 | 0.938 | 0.923 | 0.891 | 0.984 | 0.930 | 0.930 | 0.876 |
| Sub6 | 0.982 | 0.946 | 0.929 | 0.876 | 0.974 | 0.964 | 0.929 | 0.867 |
| Sub7 | 0.985 | 0.977 | 0.977 | 0.955 | 0.977 | 0.985 | 0.977 | 0.948 |
| Sub8 | 0.985 | 0.955 | 0.947 | 0.939 | 0.985 | 0.939 | 0.917 | 0.939 |
| Sub9 | 0.966 | 0.914 | 0.922 | 0.871 | 0.957 | 0.905 | 0.922 | 0.845 |
| avg | 0.970 | 0.949 | 0.938 | 0.912 | 0.976 | 0.945 | 0.930 | 0.906 |
| std | 0.013 | 0.022 | 0.017 | 0.034 | 0.011 | 0.022 | 0.023 | 0.038 |

**Table 2.** The validation accuracy for AS algorithm.

| | 6 Windows | | | | 11 Windows | | | |
|---|---|---|---|---|---|---|---|---|
| | 1 s | 1.5 s | 2 s | 2.5 s | 1 s | 1.5 s | 2 s | 2.5 s |
| Sub1 | 0.971 | 0.956 | 0.913 | 0.913 | 0.971 | 0.942 | 0.927 | 0.913 |
| Sub2 | 0.971 | 0.971 | 0.956 | 0.912 | 0.971 | 0.971 | 0.941 | 0.853 |
| Sub3 | 0.971 | 0.941 | 0.942 | 0.912 | 0.971 | 0.956 | 0.898 | 0.898 |
| Sub4 | 0.969 | 0.954 | 0.938 | 0.953 | 0.969 | 0.954 | 0.953 | 0.923 |
| Sub5 | 0.953 | 0.938 | 0.954 | 0.938 | 0.953 | 0.922 | 0.938 | 0.892 |
| Sub6 | 0.970 | 0.947 | 0.912 | 0.876 | 0.970 | 0.965 | 0.894 | 0.876 |
| Sub7 | 0.970 | 0.970 | 0.955 | 0.970 | 0.970 | 0.955 | 0.955 | 0.955 |
| Sub8 | 0.970 | 0.955 | 0.970 | 0.924 | 0.970 | 0.955 | 0.955 | 0.924 |
| Sub9 | 0.966 | 0.948 | 0.931 | 0.897 | 0.948 | 0.914 | 0.897 | 0.897 |
| avg | 0.968 | 0.953 | 0.941 | 0.922 | 0.966 | 0.948 | 0.929 | 0.903 |
| std | 0.005 | 0.010 | 0.019 | 0.027 | 0.008 | 0.018 | 0.024 | 0.028 |

The number of windows (n), step length (s) of the sliding windows, and the length of the window (w) are the parameters for the proposed MTF-CSP algorithm. The influence of these parameters on the feature extraction performance is investigated in this section, as shown in Table 3. The proposed methods with different window lengths (1 s, 1.5 s, 2 s, 2.5 s) and different numbers of windows (6, 11 windows) are compared, and two decision recognition algorithms (ED, AS) are used as the decision algorithm to make a comparison.

**Table 3.** The parameter configuration of the experiment.

| Parameters | Value |
|---|---|
| n | 6/11 |
| w | 1 s/1.5 s/2 s/2.5 s |
| m | 2 |

Figures 8 and 9 show the average classification accuracy and AUC values (expressed as a bar) and their standard deviation (expressed as a black line segment attached in the middle of the bar) for nine subjects using our proposed method in cases where window numbers, window length and decision algorithms are difference. The standard deviation is used to measure the stability of the model across different subjects. Figures 8 and 9 show the results for six and eleven windows, respectively.

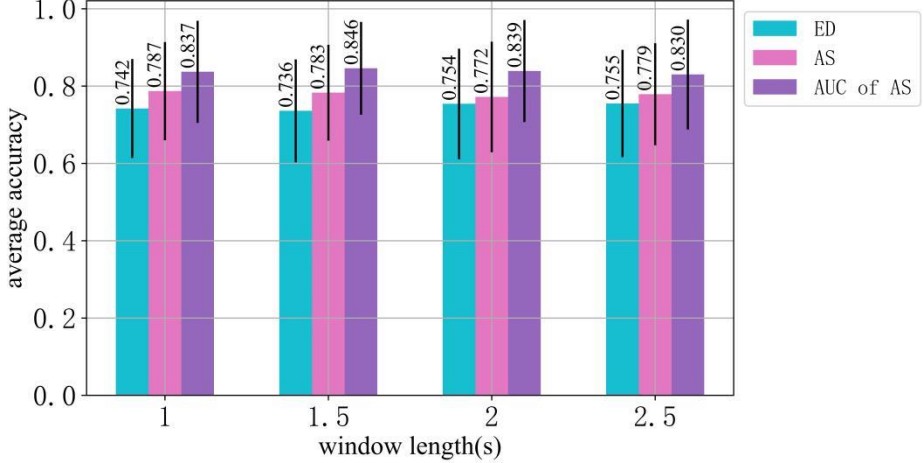

**Figure 8.** The comparison of the average classification accuracy and AUC value distribution using the ED and AS algorithms for the features extracted from six cropped windows (n = 6), each of which is 1 s/1.5 s/2 s/2.5 s in length.

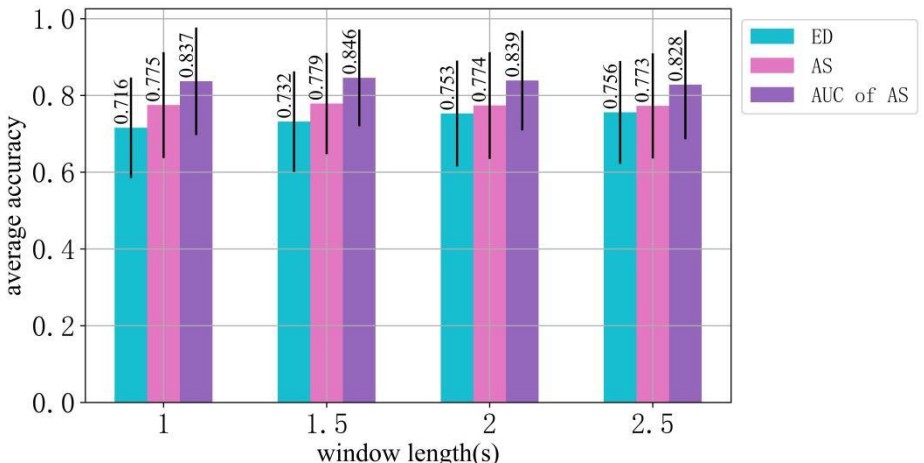

**Figure 9.** The comparison of the average classification accuracy and AUC value distribution using the ED and AS algorithms for the features extracted from 11 cropped windows (n = 11), each of which is 1 s/1.5 s/2 s/2.5 s in length.

For the ED algorithm, the performance of the features with a longer time window was better than it was with a shorter time window. In contrast, for the AS algorithm, the performance of the features with a shorter time window was better than it was with a longer time window. Generally, the average accuracy of the AS algorithm is higher than that of the ED algorithm, providing the optimal average accuracy of 0.787 in the case of the six 1 s-windows.

The threshold influences the performance of the AS algorithm. Figure 10 shows the ROC curves of the MTF-CSP feature extraction and AS algorithm-based ensemble SVM classifier with 6 1 s-windows. The x-coordinate represents the FP ratio, and the y-coordinate represents the TP ratio. The closer the ROC curve is to the upper left, the better the recognition performance is. The area under the curve is defined as the AUC value, and the closer the value is to 1, the better the performance is. As we can see from the ROC curves that most of the subjects demonstrated good performance in addition to the sub2, sub4, and sub6, which may be due to the promiscuous feature distribution in different categories, as referred to in Figure 4.

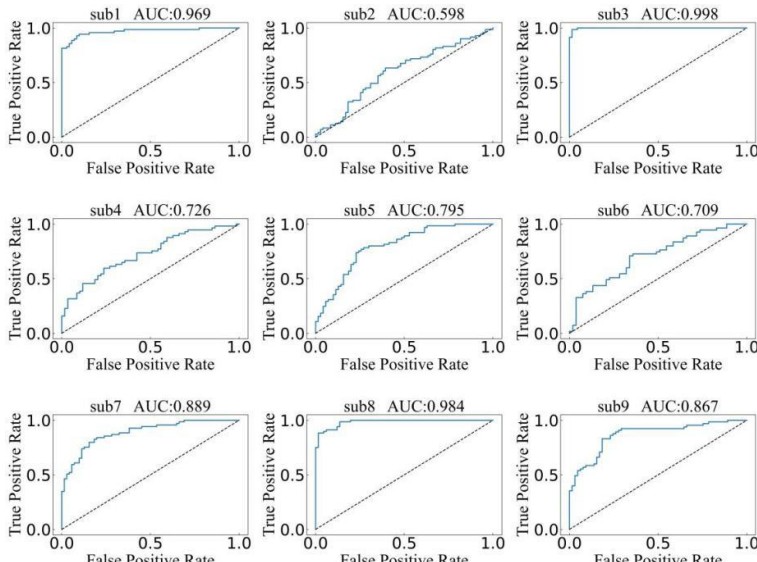

**Figure 10.** Comparison of ROC curves and the corresponding AUC values on nine subjects using the AS algorithm (The blue curve represents the ROC curve, and the black dotted line is the reference auxiliary line).

## 4. Discussion

In order to validate the superiority of the proposed MTF-CSP feature extraction, the baseline methods were compared. Figure 11 compares the proposed MTF-CSP method with the traditional CSP-based models (CSP + SVM, CSP + LDA) in terms of classification accuracy. As shown in Figure 11, the accuracy of the MTF-CSP model based on an AS algorithm and an ED algorithm is both higher than the compared models. Additionally, the proposed MTF-CSP model provides stable results with relatively lower standard deviations.

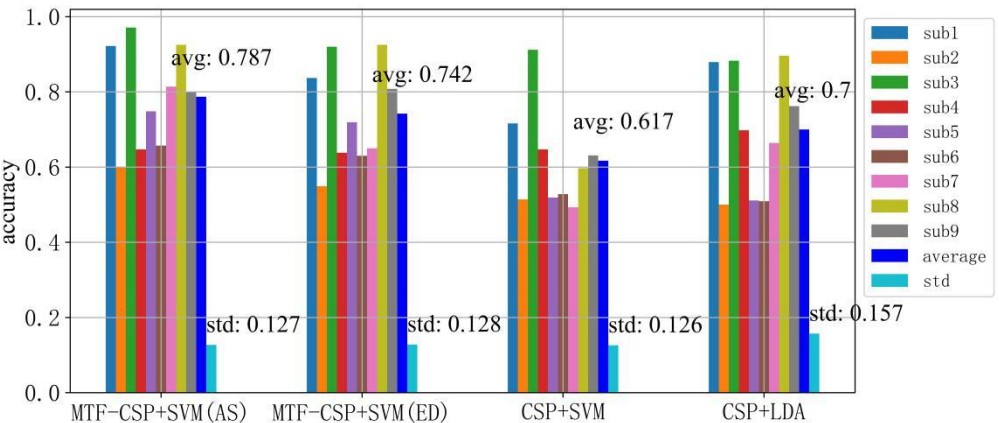

**Figure 11.** Comparison of classification accuracies of the proposed MTF-CSP based models with the traditional CSP-based models.

Several studies have attempted to create synthetic EEG time–frequency maps to reflect the time–frequency MI-EEG characteristics. As shown in Table 4, typical time–frequency maps and original signals without feature extraction are used as input features to be fed into the typical LDA and SVM models to compare them with our proposed MTF-CSP method. Additionally, the performance of the full-window multi-frequency characteristics was compared to show the effectiveness of the multi-time window strategy. As seen in Table 4, it is obvious that the model using the MTF-CSP features had better performance than the models using the original signal and time–frequency features as inputs.

**Table 4.** Comparison of classification accuracy of original timing signals, time–frequency features, and MTF-CSP features.

| Features Type | Original Sequence Signal | | STFT Time-Frequency | | Full-Window Multi-Band CSP Features | Proposed MTF-CSP Features Based on 11 2.5 s-Windows | | Proposed MTF-CSP Features Based on 6 1 s-Windows | |
|---|---|---|---|---|---|---|---|---|---|
| Classifier | LDA | SVM | LDA | SVM | SVM | ED + SVM | AS + SVM | ED + SVM | AS + SVM |
| sub1 | 0.461 | 0.511 | 0.461 | 0.468 | 0.908 | 0.837 | 0.879 | 0.837 | 0.922 |
| sub2 | 0.542 | 0.521 | 0.493 | 0.542 | 0.549 | 0.563 | 0.577 | 0.549 | 0.599 |
| sub3 | 0.460 | 0.482 | 0.686 | 0.788 | 0.964 | 0.942 | 0.971 | 0.920 | 0.971 |
| sub4 | 0.483 | 0.578 | 0.621 | 0.509 | 0.526 | 0.664 | 0.716 | 0.638 | 0.647 |
| sub5 | 0.563 | 0.674 | 0.504 | 0.533 | 0.696 | 0.659 | 0.696 | 0.719 | 0.748 |
| sub6 | 0.556 | 0.583 | 0.546 | 0.565 | 0.611 | 0.593 | 0.565 | 0.630 | 0.657 |
| sub7 | 0.736 | 0.836 | 0.521 | 0.514 | 0.779 | 0.821 | 0.814 | 0.650 | 0.814 |
| sub8 | 0.500 | 0.567 | 0.545 | 0.537 | 0.925 | 0.940 | 0.925 | 0.925 | 0.925 |
| sub9 | 0.523 | 0.562 | 0.731 | 0.831 | 0.800 | 0.785 | 0.815 | 0.808 | 0.800 |
| Avg | 0.536 | 0.590 | 0.568 | 0.587 | 0.751 | 0.756 | 0.773 | 0.742 | 0.787 |
| Std | 0.079 | 0.101 | 0.087 | 0.122 | 0.155 | 0.134 | 0.137 | 0.128 | 0.127 |

In addition to the comparison across models, we compared the average classification accuracy of the AS decision algorithm with the existing Max Voting decision algorithm, as demonstrated in Figures 12 and 13. The Max Voting algorithm works by finding the most frequent values in the sequence [44]. It can be clearly seen that the cross-session

classification accuracy obtained by the AS decision algorithm is higher than that obtained by the Max Voting algorithm in all cases.

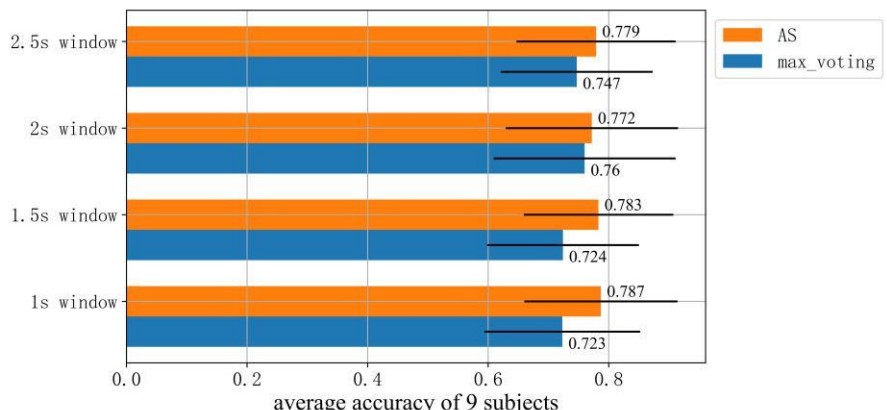

**Figure 12.** Comparison of classification accuracies of the proposed AS algorithm with the Max Voting algorithm in the case of six windows.

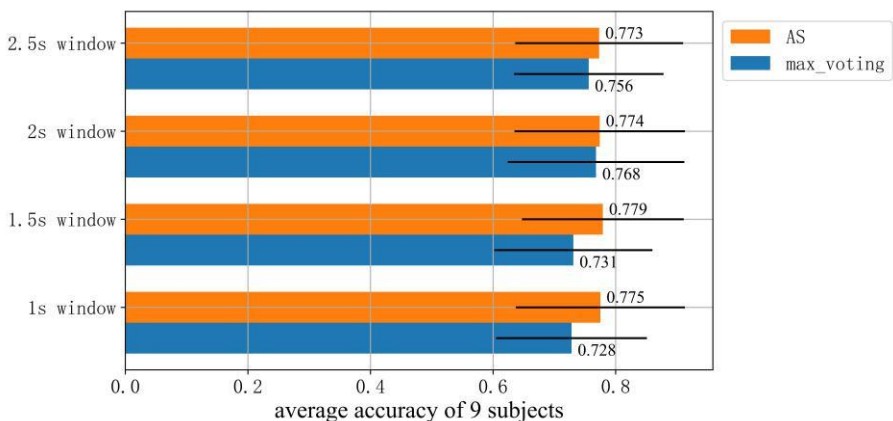

**Figure 13.** Comparison of classification accuracies of the proposed AS algorithm with the Max Voting algorithm in the case of eleven windows.

Although this study extracted EEG motor imagery features from different angles by proposing the MTF-CSP and obtained a better recognition cross-session motor imagery EEG ability than the existing methods. It has not been integrated an end-to-end system and is inconvenient for practical applications. We will try to study the current model in some end-to-end models and will apply these data to our MI-EEG decoding work.

## 5. Conclusions

This paper proposes an MTF-CSP model to extract multi-frequency bands and multi-time windows features. The extracted features were fed into the ensembled SVM classifier as an input. ED and AS algorithms were additionally proposed to obtain a final decision. The experimental results verify that a multi-frequency feature extraction strategy can refine the band information, and the multi-time windows feature extraction strategy were able to refine the temporal characteristics of the MI-EEG and caught the effective characteristics throughout the entire MI trial, significantly improving the decoding accuracy. Additionally, the strategy of the AS and ED algorithm-based ensembled SVM captured the most beneficial temporal characteristics from different angles and acquired the optimal decision result. The proposed MTF-CSP method achieves higher classification accuracy than traditional CSP + SVM and CSP + LDA models according to the present experiment. The AS and ED algorithms we propose here were inspired by the existing Max Voting algorithm and were developed based on it, adding more detailed consideration to the factors that influence the

final decision outcome from different aspects. The ED algorithm is an improved algorithm based on the max_voting algorithm by additionally considering the continuity of the same identification results rather than only focusing on the number of identical identification results. The AS algorithm was improved based on the max_voting algorithm through the more refined score values to determine the final decision rather than using crude binary classification values alone to make the final decision. According to the experiment conducted here, the proposed AS decision algorithm that was developed based on the existing Max Voting algorithm is obviously superior. Further, we determined that the AS algorithm provided a significantly better performance during the experiments than the ED algorithm did when making the final decision.

**Author Contributions:** Formal analysis, T.S.; funding acquisition, J.Y. and T.S.; investigation, J.Y. and Z.M.; methodology, Z.M.; resources, J.Y. and T.S.; software, T.S.; supervision, J.Y. and T.S.; validation, Z.M. and T.S.; visualization, Z.M.; writing—original draft, Z.M.; writing—review and editing, J.Y. All authors have read and agreed to the published version of the manuscript.

**Funding:** This research was funded by Yunnan Young Top Talents of Ten Thousand Plan (Shen Tao, Zhu Yan, Yunren Social Development No. 2018 73), grant number 2018 73, and the Introduction of Talent Research Startup Fund Project of Kunming University of Science and Technology, grant number KKSY201903028.

**Institutional Review Board Statement:** The study was conducted according to the guidelines of the Declaration of Helsinki and was approved by the Institutional Review Board of Department of Information and Automation, Kunming University of Science and Technology.

**Informed Consent Statement:** Informed consent was obtained from all subjects involved in the study.

**Data Availability Statement:** The data presented in this study are openly available in BCI Competition IV-2a: http://www.bbci.de/competition/iv/, accessed on 10 October 2021.

**Acknowledgments:** The research was totally supported and sponsored by the following project: Yunnan Young Top Talents of Ten Thousand Plan (Shen Tao, Zhu Yan, Yunren Social Development No. 2018 73), and the Introduction of Talent Research Startup Fund Project of Kunming University of Science and Technology under Program Approval Number KKSY201903028. Furthermore, we are grateful for the teachers from the Department of Information and Automation, Kunming University of Science and Technology, especially Jun Yang, Tao Shen, among others.

**Conflicts of Interest:** The authors declare no conflict of interest.

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
