# Peer review of "Multi-Time and Multi-Band CSP Motor Imagery EEG Feature Classification Algorithm"

_applsci, doi:10.3390/app112110294_

Round 1
Reviewer 1 Report
This study aimed to utilize Multi-Time and -Frequency band Common Space Pattern algorithm to classify the Motor Imagery using EEG. I have the following suggestions.
- Please add a paragraph about the contribution of this article in a bulleted form at the end part of the Introduction section.
- There are serious grammatical errors in this article, especially in the abstract. I recommend improving the grammar.
- Machine-learning-based EEG-based classification studies should be explored in the Introduction section. References should be improved by adding related articles in disease classification and prognostics, mentioning the references, such as doi.org/10.3390/brainsci11070900, /doi.org/10.1109/ACCESS.2021.3109806, doi.org/10.1109/ACCESS.2020.3040437.
- The authors addressed seven EEG frequency bands in this study. Authors should mention the names of the EEG frequency bands.
- The image quality of the Figures should be improved.
- Authors should mention the features which they extract from the frequency bands (i.e. mean power, median power, spectral edge?).
- Authors should show the statistical results of EEG features for different motor imaginary states.
- Cross-validation methodology and results should be mentioned.
- Authors must discuss the advantages and drawbacks of their proposed method with other studies adding a discussion section.
Author Response
Response to Reviewer 1 Comments
Point 1:
Please add a paragraph about the contribution of this article in a bulleted form at the end part of the Introduction section.
Response 1: we have added and prominent this part at the end part of the Introduction section.
Point 2:
There are serious grammatical errors in this article, especially in the abstract. I recommend improving the grammar.
Response 2: we have tried our best to improve our grammar, and the change records are marked by ‘Track Changes’.
Point 3:
Machine-learning-based EEG-based classification studies should be explored in the Introduction section. References should be improved by adding related articles in disease classification and prognostics, mentioning the references, such as doi.org/10.3390/brainsci11070900, /doi.org/10.1109/ACCESS.2021.3109806, doi.org/10.1109/ACCESS.2020.3040437.
Response 3: we have added this part at the second paragraph of the introduction and marked it by ‘Track Changes’. The article in disease classification and prognostics mentioned above is quoted in [16][17][18]
Point 4:
The authors addressed seven EEG frequency bands in this study. Authors should mention the names of the EEG frequency bands.
Response 4: we have named this frequency band at the first paragraph of section 2.2.2 and the naming of frequency bands in Figure 3 has been modified.
Point 5:
The image quality of the Figures should be improved.
Response 5: We have rearranged the colors, drawing methods and pixels of figures 5, 6 and 9 .
Point 6:
Authors should mention the features which they extract from the frequency bands (i.e. mean power, median power, spectral edge?).
Response6: Before input to CSP, the multi-band features are filtered from the original signal by bandpass filter, which is still a time-amplitude sequence. After feature extraction by CSP, the signals of each frequency band are mapped into a vector with a length of 2m, and the CSP features of 7 frequency bands are connected end to end to form a vector with a length of 2m*7. Your confusion may be because we have not clearly described it. For this, we have added the description of the features of the extracted multi-frequency band in the first paragraph of Section 2.2.2
Point 7:
Authors should show the statistical results of EEG features for different motor imaginary states.
Response 7: We show the different classes of multi-band and multi-period statistical features in the form of scatter plots in figures 4 .
Point 8:
Cross-validation methodology and results should be mentioned.
Response 8: we have mentioned the Cross-validation methodology at the first paragraph of section 3, and showed the Cross-validation result in the Table 2 and Table 3.
Point 9:
Authors must discuss the advantages and drawbacks of their proposed method with other studies adding a discussion section.
Response 9: at section 4,we add another discussion section to compare the proposed method with other methods. In this section, we have additionally added a comparison algorithm called Max Voting to the proposed AS algorithm,highlighting the advantages of our proposed method from the comparison results. Meanwhile, the drawbacks of our proposed method are summarized in the last paragraph of section 4.

Reviewer 2 Report
Title: “Multi-time and Multi-band CSP Motor Imagery EEG feature classification algorithm”
In this work the authors proposed a Multi-Time and -Frequency band Common Space Pattern (MTF-CSP) based feature extraction and EEG decoding method. The proposed method has been evaluated on the EEG data of 9 subjects in the 2008 BCI-2a competition dataset.
General comment: This work should be deeply reworked to provide a better main text together with a clear logical flow. Although the main aim may be clear, the assessment of the goodness of the proposed strategy should be quantitatively shown. The improvement with respect to standard procedures should be better presented with the help of numbers. Another issues is due to the number of subjects, which seems to be quite small. Is this number really adequate to be statistically significant ?Are further experiments needed ? The logic flow of the main text is perhaps difficult to follow since some parts of the text of definitely not clear. Several captions of figures should be improved and the quality of the images should be improved together with the use of labels.
Author Response
Response to Reviewer 2 Comments
Point 1:
This work should be deeply reworked to provide a better main text together with a clear logical flow.
Response 1: we have added a main text at the first paragraph of section 2.2 together with a clear logical flow in Figure 2.
Point 2:
Although the main aim may be clear, the assessment of the goodness of the proposed strategy should be quantitatively shown. The improvement with respect to standard procedures should be better presented with the help of numbers.
Response 2: At the section 3, we have give a lot of clear quantitatively validation and test accuracy results of the proposed method. The ROC curves of each subject have also been shown in Figure 10 to represent the classification performance. Aditionally, we have add another discussion section (section 4) where we compare our metonds with other methods from different aspects in forms of statistical bar chart, In this section, we have additionally added a comparative algorithm called Max Voting to the proposed AS algorithm, highlighting the advantages of our proposed method from the comparison results.
Point 3:
Another issues is due to the number of subjects, which seems to be quite small. Is this number really adequate to be statistically significant ?Are further experiments needed ?
Response 3: There are still some issues to consider when it comes to the lack of data:
First of all, there is an objective reason that the time limit of this modification is 10 days. It takes a lot of time to find suitable data sets and complete all series of experiments, so it is impossible to complete all experiments in such a short time;
Second, in the latest available publicly datasets, there is a limited number of datasets that carry multiple channels of information without adding other redundant components , some datasets, while recording the motor imagery signal, but the experimental paradigm is incorporating the feedback process, which makes the EEG signals were collected with redundant components, and is not consistent with our research, We have selected datasets from the existing public datasets that have a large amount of data collected per subject, sufficient number of channels and relatively pure motor imagination EEG signals as experimental objects, so as to measure the model from a more accurate perspective. According to different problems, the existing and the appropriate data is limited, this is a common problem, so a lot of research is also use only one dataset to verify their models, such as https://doi.org/10.1109/TNSRE.2020.3027004.
Indeed, your suggestion is of great significance to our future research, we will also try to generalize the method to more different types of datasets, and try to use our own devices to collect private data for model validation combined with public datasets in the future research.
Point 4:
The logic flow of the main text is perhaps difficult to follow since some parts of the text of definitely not clear.
Response 4: We revised some of the expressions in the article to make the logic more rigorous and marked it by ’Track Changes’.
Point 5:
Several captions of figures should be improved and the quality of the images should be improved together with the use of labels.
Response 5: We have improved the captions and picture quality of all figures.

Reviewer 3 Report
First of all, the paper should be improved in terms of language with the help of an English speaker and paying attention to typos (especially on the punctuation). I also suggest to reshape the paper structure to follow the typical sequence from subjects to discussion, and then the results.
Secondly, the information on the subjects should be extended (age and characteristics of the sample, why was this sample size adopted?). I have some concerns on the approval of an institutional committee instead of an ethical committee for evaluating a biosignal-collection protocol, but it's a minor because I simply don't know the legal context in the country where the experiments were performed.
I cannot understand how the standard deviations are represented in graphs 6, 7, 9.
Author Response
Response to Reviewer 3 Comments
Point 1:
First of all, the paper should be improved in terms of language with the help of an English speaker and paying attention to typos (especially on the punctuation). I also suggest to reshape the paper structure to follow the typical sequence from subjects to discussion, and then the results.
Response 1: We have identified spelling and grammatical errors in the article and made corrections, as well as improved the grammar, the changes are marked by ‘Track Changes’. We have add a discussion part at section 4.
Point 2:
Secondly, the information on the subjects should be extended (age and characteristics of the sample, why was this sample size adopted?). I have some concerns on the approval of an institutional committee instead of an ethical committee for evaluating a biosignal-collection protocol, but it's a minor because I simply don't know the legal context in the country where the experiments were performed.
Response 2: The data we used is from public BCI competition, according to the description document from data providers, they are not particularly illustrate the health status of the subjects and age characteristics. Due to the used dataset is already existing, and a lot of research has used this dataset, so we ignore the data acquisition protocol approval.
Point 3:
I cannot understand how the standard deviations are represented in graphs 6, 7, 9.
Response 3: The standard deviation is used to measure the stability of the model across different subjects. we have added explanation at the end of the second paragraph of section 3. The standard deviation in figure 6 and figure 7 might not be represented in the figure n a appropriate way, so we show the standard deviation as a black line in the middle of the bar, and redraw the bar chart to improve the quality of the chart, as shown in figure 8 and figure 9.

Round 2
Reviewer 2 Report
Title: “Multi-time and Multi-band CSP Motor Imagery EEG feature classification algorithm”
General comment: The authors revised this work. However, some points should be further reworked to improve the quality of the manuscript.
In particular, some parts of the main text should be made more clear to the interested readers. Similarly, some figures are still not clear and should be made more accessible to the readers by improving their quality.
Some detailed comments:
2.2.1. CSP algorithm
*) This paragraph should improved and made more clear for interested readers with a more general background and not specific background
2.2.3. SVM classifier for multi-window EEG classification
*) see the previous comment. In addition Eq(5) should be explained in a clear way as well as all the used symbols.
Please explain in details the meaning of “The radial basis function (RBF) is used as the kernel function in this paper”. What the RBF functions look like ? Why they are a basis. Please provide a clear explanation and explicit these function in the main text.
2. Average score algorithm (AS)
*) Also this paragraph is not totally clear. Please improve.
Figure 7. The precision and recall curves for nine subjects according to a different threshold
values
*) Please, improve the caption and the labels to allow the interested readers to better understand this panel.
Figure 8. The comparison of the average classification accuracy and AUC values distribution
using the ED and AS algorithms for the features extracted from 6 cropped windows (n=6), each of
which is 1s/1.5s/2s/2.5s in length.
Figure 9. The comparison of the average classification accuracy and AUC values distribution
using the ED and AS algorithms for the features extracted from 11 cropped windows (n=11) , each
of which is 1s/1.5s/2s/2.5s in length.
*)Please improve the labels in the figures 8 and 9
Figure 10. ROC curves of AS algorithm on nine subjects.
*) Please improve the caption and the labels in this figure.
Lines: “Also, the strategy of AS and ED algorithm based ensembled SVM capture the most
beneficial temporal characteristics from different angles and get the optimal decision
result. The proposed MTF-CSP method achieves higher classification accuracy than the
traditional CSP+SVM and CSP+LDA models according to the experiment. At the same
time, the proposed AS decision algorithm is obviously superior to the existing Max
Voting algorithm. Further, we found out from experiments that the AS algorithm
provides significantly better performance than the ED algorithm when making the final
decision.”
*) The value of this work should be explained in a more detailed way. Why just the Max
Voting algorithm was chosen ? Why just AS and ED were selected ? Please explain better.
Author Response
Response to Reviewer 2 Comments
Point 1:
2.2.1. CSP algorithm
*) This paragraph should improved and made more clear for interested readers with a more general background and not specific background

Response 1: We have add the general background of CSP algorithm at the beginning of section 2.2.1 in the form of ‘Track Changes’.
Point 2:
2.2.3. SVM classifier for multi-window EEG classification
*) see the previous comment. In addition Eq(5) should be explained in a clear way as well as all the used symbols.
Please explain in details the meaning of “The radial basis function (RBF) is used as the kernel function in this paper”. What the RBF functions look like ? Why they are a basis. Please provide a clear explanation and explicit these function in the main text.
Response 2: We have add the general background of SVM algorithm at the beginning of section 2.2.3 and a more detailed description of the formula 5 was added under it in the form of ‘Track Changes’.
After this, we insert Formula 6 and the associated description to explain the meaning and function of RBF.
Point 3:
- Average score algorithm (AS)
*) Also this paragraph is not totally clear. Please improve.
Response 3: We have improved this paragraph in the form of ‘Track Changes’.
Point 4:
Figure 7. The precision and recall curves for nine subjects according to a different threshold
values
*) Please, improve the caption and the labels to allow the interested readers to better understand this panel.
Response 4: We have improved the caption of Figure 7 in the form of ‘Track Changes’.
Point 5:
Figure 8. The comparison of the average classification accuracy and AUC values distribution
using the ED and AS algorithms for the features extracted from 6 cropped windows (n=6), each of
which is 1s/1.5s/2s/2.5s in length.
Figure 9. The comparison of the average classification accuracy and AUC values distribution
using the ED and AS algorithms for the features extracted from 11 cropped windows (n=11) , each
of which is 1s/1.5s/2s/2.5s in length.
*)Please improve the labels in the figures 8 and 9
Response 5: We have improved the labels in the figures 8 and 9 in the form of ‘Track Changes’
Point 6:
Figure 10. ROC curves of AS algorithm on nine subjects.
*) Please improve the caption and the labels in this figure.
Response 6: We have improved the caption and the labels in this figure in the form of ‘Track Changes’ .
Point 7:
Lines: “Also, the strategy of AS and ED algorithm based ensembled SVM capture the most
beneficial temporal characteristics from different angles and get the optimal decision
result. The proposed MTF-CSP method achieves higher classification accuracy than the
traditional CSP+SVM and CSP+LDA models according to the experiment. At the same
time, the proposed AS decision algorithm is obviously superior to the existing Max
Voting algorithm. Further, we found out from experiments that the AS algorithm
provides significantly better performance than the ED algorithm when making the final
decision.”
*) The value of this work should be explained in a more detailed way. Why just the Max
Voting algorithm was chosen ? Why just AS and ED were selected ? Please explain better.
Response 7: We have add the explanation of the value of this work in a more detailed way at the part 5 in the form of ‘Track Changes’.
